# New Approaches to Targeting Epigenetic Regulation in Bladder Cancer

**DOI:** 10.3390/cancers15061856

**Published:** 2023-03-20

**Authors:** Daryl Thompson, Nathan Lawrentschuk, Damien Bolton

**Affiliations:** 1Department of Surgery, Austin Health, The University of Melbourne, Heidelberg, VIC 3084, Australia; 2Division of Cancer Surgery, Peter MacCallum Cancer Centre, The University of Melbourne, Melbourne, VIC 3000, Australia; 3Department of Urology, The Royal Melbourne Hospital, Melbourne, VIC 3050, Australia; 4EJ Whitten Prostate Cancer Research Centre at Epworth Healthcare, Melbourne, VC 3121, Australia; 5Olivia Newton-John Cancer and Wellness Centre, Austin Health, Melbourne, VIC 3084, Australia

**Keywords:** epigenetics, epigenetic inhibitors, metastatic bladder cancer, urothelial bladder cancer

## Abstract

**Simple Summary:**

Epigenetic changes occur in parts of the genome other than in nucleotides. They are considered reversible and are therefore important targets for cancer therapy. Epigenetic changes have been observed in urological cancers, including urothelial carcinoma, and in recent years have been a topic of investigation for the treatment of metastatic bladder cancer that has failed traditional therapy. We performed a review of the current literature to assess the evidence and role for targeted epigenetic therapy in bladder cancer. While we found 25 clinical trials investigating this topic, there have been no phase 3 human clinical trials to date. This is an emerging topic in urology, and future directions involve further research into bladder cancer-specific epigenetic changes, as well as the development of novel agents to target these mutations.

**Abstract:**

Epigenetics is a growing field and in bladder cancer, it is of particular interest in advanced or metastatic disease. As opposed to genetic mutations in which the nucleotide sequence itself is altered, epigenetic alterations refer to changes to the genome that do not involve nucleotides. This is of great interest in cancer research because epigenetic alterations are reversible, making them a promising target for pharmacological agents. While chemoimmunotherapy is the mainstay for metastatic disease, there are few alternatives for patients who have progressed on first- or second-line treatment. By targeting reversible epigenetic alterations, novel epigenetic therapies are important potential treatment options for these patients. A search of clinical registries was performed in order to identify and collate epigenetic therapies currently in human trials. A literature search was also performed to identify therapies that are currently in preclinical stages, whether this be in vivo or in vitro models. Twenty-five clinical trials were identified that investigated the use of epigenetic inhibitors in patients with bladder cancer, often in combination with another agent, such as platinum-based chemotherapy or pembrolizumab. The main classes of epigenetic inhibitors studied include DNA-methyltransferase (DNMT) inhibitors, histone deacetylase (HDAC) inhibitors, and histone methyltransferase (HMT) inhibitors. At present, no phase 3 clinical trials have been registered. Few trials have published results, though DNMT inhibitors have shown the most promise thus far. Many patients with advanced or metastatic bladder cancer have limited treatment options, particularly when first- or second-line chemoimmunotherapy fails. Epigenetic alterations, which are common in bladder cancer, are potential targets for drug therapies, and these epigenetic agents are already in use for many cancers. While they have shown promise in pre-clinical trials for bladder cancer, more research is needed to assess their benefit in clinical settings.

## 1. Introduction

Bladder cancer is a common urological malignancy with approximately 573,000 new diagnoses and 213,000 deaths reported worldwide in 2020, and among men, it ranks 6th for new cancer diagnoses and 9th for cancer deaths [1]. The 5-year survival rate for bladder cancer is reported as 54%, and even after curative-intent treatment with radical cystectomy, disease recurrence ensues in up to 30% [2,3]. Advanced or metastatic urothelial cancer of the bladder carries an even poorer prognosis with an overall survival of just over one year, even with first line chemotherapy and immunotherapy [4].

The current standard of care for patients with metastatic urothelial cancer depends on fitness for platinum. Platinum-fit patients typically receive cisplatin- or carboplatin-containing chemotherapy first-line, whereas platinum-unfit patients receive checkpoint inhibitors, such as pembrolizumab [5]. Patients who fail chemoimmunotherapy, however, have limited treatment options. For this reason, several novel agents are currently being studied in both pre-clinical and clinical trials. These include epigenetic inhibitors, which are drugs that target epigenetic alterations found in multiple human cancers. Epigenetic therapy is already established in the treatment of haematological malignancies and is increasingly being studied in urological cancers. In this review, we aim to summarise the current status of epigenetics and potential epigenetic treatments for bladder cancer.

## 2. Epigenetics and Bladder Cancer

Epigenetics is a term that was first used in 1940 to broadly describe anything relating to the process by which a genotype is expressed as a phenotype [6]. Today epigenetic alterations refer to acquired or heritable changes in a gene’s function that do not involve disruption of the nucleotide sequence [7]. This is in contrast to genetic mutations, in which the nucleotide sequences themselves are affected. Epigenetic alterations have been found in a variety of human malignancies and are thought to be reversible, making them promising targets for novel cancer therapies [8]. This is of particular interest in advanced or metastatic malignancies for which limited treatment options exist.

The enzymes responsible for epigenetic alterations can be categorised into three broad groups: writers, erasers, and readers. ‘Writers’ are proteins that add modifications, such as methyl or acetyl groups, to DNA or histones [9]. Common writers implicated in human cancers include DNA methyltransferases (DNMTs), histone-lysine N-methyltransferases (HMTs), and histone acetyltransferases (HATs). ‘Erasers’ directly oppose the action of writers by catalysing the removal of acetyl or methyl groups and include histone deacetylases (HDACs) [9]. The final group, ‘readers’, are proteins that are responsible for enacting the alterations made by writers and erasers [9].

### 2.1. Writers

#### 2.1.1. DNA Methyltransferases (DNMTs)

DNA methylation, carried out by DNMTs, involves the addition of a methyl group to CpG sites, which is the region of DNA in which a cytosine nucleotide is followed by a guanine nucleotide (Figure 1) [10]. There are four types of DNMTs found in mammals, of which DNMT1 is the most abundant [11]. DNA methylation is essential for normal cell growth and development, imprinting, and X-chromosome inactivation [8]. Given that up to 80% of human CpG sites are methylated [10], any deviation from the normal process can have profound effects on gene expression. Aberrant methylation, whether it be hyper- or hypomethylation, has been observed in many cancers, particularly when it occurs in gene promoters, oncogenes, or tumour suppressor genes [8].

Patients with urothelial carcinoma of the bladder have been shown to have hypermethylation in multiple genes, with several studies even finding that the detection of hypermethylation in urine samples has higher sensitivity for diagnosing bladder cancer than traditional cytology [12,13]. There is also evidence to suggest that the degree of hypermethylation correlates with the aggressiveness of the cancer [14,15,16]. Hypermethylation in certain promoters has also been linked to tobacco smoking, a known risk factor for bladder cancer [15]. Tissue studies, such as that performed by Liu et al., have shown that DNMT1 is upregulated in bladder cancer samples compared to levels in the normal urothelium [17]. Here, the authors showed that silencing DNMT1 inhibited the growth and migration of tumour cells, whereas increasing DNMT1 expression had the opposite effect, confirming its role in bladder cancer [17].

#### 2.1.2. Histone-Lysine N-Methyltransferases (HMTs)

HMTs are enzymes that catalyse the addition of methyl groups to histones, which are the proteins around which DNA is wound into nucleosomes [18]. Like DNA methylation, histone methylation also plays a crucial role in normal development and is involved in processes, including DNA replication and repair and gene transcription [19]. Aberrant histone methylation has been observed in many human cancers, and the most well-known aberrations occur in enhancer of zeste 2 (EZH2), which is thought to be the main enzyme involved in histone modification [20].

In urological oncology EZH2 is best known for its role in castrate-resistant prostate cancer (CRPC), in which its mutated form can activate genes involved in the androgen receptor pathway [21]. It has also been implicated in bladder cancer, with one study finding a correlation between EZH2 overexpression and non-muscle invasive bladder cancer (NMIBC) in both mouse models and humans, as well as an increased likelihood of disease recurrence in those with EZH2 overexpression [22]. Warrick et al. found that EZH2 expression was highest in bladder CIS, followed in descending order by muscle invasive bladder cancer (MIBC), NMIBC, and the normal epithelium, though they did not find an association with the oncological outcome [23].

Similar findings were observed by Chen et al., leading authors to conclude that not only is EZH2 associated with bladder cancer, but that its degree of expression correlates with disease severity [24]. EZH2 has also been found in urine through RNA released from cancer cells, and Zhang et al. report that its measurement can distinguish between MIBC and NMIBC and is a more sensitive diagnostic test than urine cytology [25].

#### 2.1.3. Histone Acetyltransferases (HATs)

HATs also act on histones but catalyse the addition of acetyl groups rather than methyl groups. They are a diverse group of enzymes that play an important role in DNA repair and overall gene stability [26]. One protein complex demonstrating HAT activity, though not technically classified as a HAT, is the CREB-binding protein (CBP)/p300 coactivator family. CBP/p300 acetylates various oncoproteins and tumour-suppressor proteins, and thus, it is via these relationships that CBP/p300 abnormalities are associated with cancer [26].

While CBP/p300 mutations are more closely associated in haematological malignancies, they have also been found in many solid tumours, including colorectal, breast, ovarian, and pancreatic cancers [27]. Its impact on the androgen receptor pathway and subsequent association with prostate cancer is well known [28,29], though its role in bladder cancer is less clear. A 2011 study, however, noted that p300 is under-expressed in doxorubicin-resistant bladder cancer cells, generating interest in this class as a potential therapeutic target for chemotherapy-resistant cancers [30].

### 2.2. Erasers

#### 2.2.1. Histone Deacetylases (HDACs)

HDACs are a group of enzymes that oppose the actions of HATs by catalysing the removal of acetyl groups from histones. This is a large group comprising 18 subtypes in humans, which are further divided into four classes [31].

Class I enzymes, which include HDAC1, 2, 3, and 8, are expressed in all cells, whereas the other three classes are more tissue-specific [32]. Just like their HAT counterparts, HDACs act on histones to regulate gene expression, apoptosis, and cell proliferation [31]. Inappropriate expression of these enzymes has been identified in haematological malignancies, such as acute myeloid leukaemia (AML) and non-Hodgkin’s lymphoma, as well in a range of solid tumours, including breast, cervical, colon, and prostate cancers [33].

More is known about the relationship between bladder cancer and class I HDACs, though enzymes from other classes have been studied as well [32]. Mutations in multiple HDAC genes have been identified in urothelial bladder cancers, leading to the overexpression of HDAC1 [34], HDAC2, and HDAC8, as well as the under-expression of HDAC5 and HDAC7 [35]. HDAC4’s association with bladder cancer is somewhat unclear, with one study finding it under-expressed in urothelial cancer [35] and another showing it as both overexpressed and associated with more severe disease [36]. Aberrations in HDAC1, in particular, are associated with poorer prognosis compared to other HDAC types [37].

HDAC3 microRNA in urine has also been shown to be significantly higher in patients with urothelial bladder cancer than in control patients [38]. HDACs have also been studied in non-urothelial bladder cancer, with one study finding HDAC4, HDAC7, and HDAC9 to be overexpressed in basal-squamous bladder cancer [39].

#### 2.2.2. DNA Demethylases

DNA demethylases counteract DNMTs by removing a methyl group from the CpG sites. DNA hypomethylation was actually the first epigenetic alteration implicated in tumour development, though it was quickly overlooked as researchers shifted their focus to hypermethylation. Nonetheless, hypomethylation is a feature of several cancers, including ovarian epithelial carcinoma, metastatic prostate cancer, hepatocellular carcinoma, and colon adenocarcinoma [40]. In bladder cancer, hypomethylation has not been studied as thoroughly as hypermethylation, and most current research focuses on inducing hypomethylation to slow or reverse tumour growth. However, the upregulation of DNA demethylase has been noted in bladder cancer, although it appears to be linked to low-grade tumours, whereas hypermethylation is more common in high grade tumours [41].

#### 2.2.3. Histone Demethylases (HDMs)

The addition of methyl groups to histones by methyltransferases was historically thought to be an irreversible process, and it was not until 2004 that the discovery of the first histone demethylase demonstrated that methyl groups could in fact be removed [42]. Histone demethylases (HDMs) are grouped by their mechanism of action and comprise two families: lysine-specific demethylases (LSDs) and Jumonji C-domain-containing demethylases (JMJDs), both of which are overexpressed in certain malignancies, such as prostate cancer [43]. While JMJDs appear protective against bladder cancer, LSD1 has been reported at higher levels in urothelial cancer compared to those in the normal urothelium [44]. Evidence on LSD1’s prognostic value is mixed, however, with some studies finding it linked to low grade but not high grade tumours [44], and others finding it significantly overexpressed in high-grade or metastatic cancers [45].

### 2.3. Readers

#### 2.3.1. Bromodomain and Extraterminal Domain (BET)

The bromodomain and extraterminal domain (BET) family is a group of proteins that includes BRD2, BRD3, BRD4, and BRDT. While each protein performs a different function, they are all involved in the recognition of acetylated lysine residues on histones [46]. By ‘reading’ acetyl groups put down by HATs, BET proteins can regulate the activity and function of enzymes and proteins, thereby affecting gene transcription and chromatin remodelling [47].

One such type of gene directly regulated by BET proteins is the Myc family, which includes regulator genes and proto-oncogenes. Abnormal BET activity can result in the amplification or overexpression of Myc genes, resulting in tumorigenesis [48]. Myc genes are major players in haematological malignancies, such as Burkitt lymphoma, but are also important in many solid tumours, including neuroblastoma, melanoma, and breast cancer [48].

BRD4 is particularly relevant in urothelial bladder cancer and has been shown to be overexpressed in cancerous cells compared to levels in the normal urothelium, likely causing tumorigenesis through the Myc pathway [49]. Higher BRD4 expression has also been linked to a higher histological grade, as well as lymph node and distant metastasis [50].

#### 2.3.2. Methyl-CpG-Binding Domain (MBD) Proteins

MBD proteins are a group of readers that are involved in the readout of DNA methylation, though they may also influence chromatin remodelling and histone methylation [51]. They are most well-known for their role in Rett syndrome, a neurodegenerative disorder caused by an X-linked mutation of the MBD gene *MECP2*, in which the gene loses its function [52]. Loss-of-function mutations in MBD protein genes have also been reported in human malignancies, including breast, lung, pancreas, colon, and prostate cancers [51]. Because MBD proteins help regulate DNA methylation, the absence of these proteins can cause DNA methylation to occur unchecked, which can be cancer-inducing as discussed previously. MBD2, a type of MBD protein, is thought to have a protective role against bladder cancer as demonstrated by Zhu et al., whose 2004 study reported that high expression of MBD2 is associated with a reduced risk of urothelial carcinoma [53].

## 3. Pre-Clinical and Clinical Trials

### 3.1. DNA Methyltransferases (DNMTs)

DNMT inhibitors are one of the few classes of epigenetic inhibitors approved by the American Food and Drug Association (FDA) for use in malignancy. These approved agents, azacytidine and decitabine, are currently only approved for use in myelodysplastic syndrome, though they have also been trialled for other malignancies, including breast and prostate [54]. Azacytidine acts on all DNMT classes, whereas decitabine acts preferentially on DNMT1.

To date, eight clinical trials have been performed using DNMT inhibitors in patients with advanced or metastatic urothelial bladder cancer, often in combination with platinum-based chemotherapy or pembrolizumab (Table 1). Three of these studies were phase I dose-finding trials, reporting that azacytidine and the novel agent guadecitabine were well-tolerated [55,56,57]. Only one phase 2 trial has been completed, and although it was terminated early due to slow accrual, progression-free survival was better than expected in urothelial cancer patients treated with the DNMT inhibitor 5-fluoro-2′-deoxycytidine [58]. There is currently one active phase 2 trial investigating guadecitabine in combination with atezolizumab, and it was expected to be completed in 2022 (NCT03179943). A phase 1/2 trial using NTX-301, an inhibitor targeting DNMT1 specifically, in combination with platinum-based chemotherapy, is currently recruiting (NCT04851834). A further studying using azacytidine was terminated early by sponsors, though further details were not provided (NCT02959437).

While human clinical trials have thus far administered azacytidine as a subcutaneous or intravenous injection, intravesical instillation has recently been investigated in a 2021 study using rat models, which reported a reduction in tumour burden using this method [61]. Other DNMT inhibitors that have not yet progressed to clinical trials include chromobox 7 (CBX7), which has been shown to cause tumour suppression in vitro [62], and CM-272, which can cause tumour and metastasis regression in in vivo transgenic mouse models [63].

### 3.2. Histone Methyltransferases (HMTs)

Inhibition of Enhancer of Zeste 2 (EZH2), a type of HMT, has been proven effective for metastatic or locally advanced epithelioid sarcoma and it is for this indication that the EZH2 inhibitor tazemetostat has been approved by the FDA [54]. Other EZH2 inhibitors are undergoing clinical trials to assess efficacy in myeloma, non-Hodgkin’s and B-cell lymphoma, and solid tumours, including breast, lung, and prostate, though none have progressed to phase 3 trials [64].

Few clinical trials have been conducted using EZH2 inhibitors in bladder cancer (Table 1), and the single completed trial has not reported results (NCT03525795). A phase 1/2 trial investigating tazemetostat in combination with pembrolizumab is currently recruiting patients with locally advanced or metastatic bladder cancer who have progressed on first-line chemotherapy (NCT03854474).

Despite the paucity of human clinical trials, there have been multiple studies published on EZH2 inhibitors using in vitro cell lines or animal models. The EZH2 inhibitor EPZ011989 has been reported to cause cell cycle arrest in in vitro assays and decreased tumour volume in xenograft mouse models [65]. UNC1999 is another EZH2 inhibitor that is observed to have an anti-tumorigenic effect on human bladder cancer cells [24]. EZH2 is also thought to have a direct relationship with lysine-specific demethylase 6A (LSD-6A), which is a histone demethylase and therefore performs the opposite function of EZH2. Several studies have reported that bladder cancers showing mutations in *KDM6A*, the gene coding for LSD-6A, are especially susceptible to EZH2 inhibition [65,66]. This is an interesting prospect given the rise of personal genomics as it suggests that particular genetic mutations may be more amenable to one kind of treatment than another.

### 3.3. Histone Acetyltransferases (HATs)

The inhibition of HAT-like CBP/p300 in cancer is less well-studied than other epigenetic targets with currently no approved agents in use. Clinical trials are also lacking with only three active currently and none completed. CCS1477 is a CBP/p300 inhibitor with phase 1/2 trials currently recruiting patients with haematological malignancy (NCT04068597), as well as advanced or metastatic solid tumours, including breast, lung, and prostate (NCT03568656). A phase 1 clinical trial using another CBP/p300 inhibitor, FT-7051, is also currently recruiting and is focusing on patients with metastatic CRPC (NCT04575766). To date, there are no past or current trials using CBP/p300 inhibitors in bladder cancer patients.

There is also a paucity of pre-clinical data investigating the use of CBP/p300 inhibitors in bladder cancer. A 2019 in vitro study had promising results, finding that CBP/p300 inhibition in bladder cancer cells led to decreased Myc expression, thereby increasing apoptosis and reducing the proliferation of malignant cells [67]. However, CBP/p300 inhibition in this study was performed using the process of transfection rather than via the use of an epigenetic agent. Thus, further studies are needed to identify CBP/p300-inhibiting drugs and to test their effects on urothelial cancer.

### 3.4. Histone Deacetylases (HDACs)

HDAC inhibitors are a well-studied group and make up three of the seven currently approved epigenetic agents, all of which are in use for T-cell lymphoma [54]. Vorinostat, also known as suberoylanilide hydroxamic acid or SAHA, was the first epigenetic agent approved by the FDA and is used mainly for refractory cutaneous T-cell lymphoma (CTCL) [54]. It also has shown an effect in clinical trials investigating lymphoma, lung cancer, breast cancer, head and neck carcinoma, and colorectal and prostate cancers [66]. As an HDAC inhibitor, its primary effect is on acetylation, but in vitro studies have shown that it may also have some action on methylation and miRNA expression in certain cancers [66].

Thus far, vorinostat has not yielded particularly promising results in human bladder cancer trials. A phase 1 trial combining vorinostat with docetaxel was terminated in 2008 due to toxicity (NCT00565227), and a phase 2 trial using it as a single agent was terminated due to both a lack of demonstrable efficacy, as well as intolerable side effects [59]. One trial using vorinostat as a single agent ran to completion in 2008, though results have not been published (NCT00045006). There is currently one active trial investigating vorinostat in combination with pembrolizumab in patients with metastatic bladder cancer who have failed chemotherapy, which was expected to finish in 2022 (NCT02619253).

Belinostat, or PXD101, is another FDA-approved HDAC inhibitor that is used for peripheral T-cell lymphoma (PTCL) [54]. It has also progressed to phase 2 trials for ovarian cancer [68], hepatocellular carcinoma [69], and AML [70]. In vivo urothelial cancer trials using animal models have reported promising results, including a reduced tumour volume, less haematuria, and slower disease progression, in mice treated with belinostat [71,72]. There have been three completed clinical trials using belinostat in patients with refractory bladder cancer, two of which have not reported results (NCT00413075, NCT00413322). The third, which combined belinostat with carboplatin or paclitaxel, has not published results but preliminary outcomes show at least a partial response in four of fifteen enrolled patients (NCT00421889). A further phase 1 trial investigating belinostat in combination with tremelimumab and durvalumab is currently recruiting (NCT05154994).

The third HDAC inhibitor that has been granted FDA approval is FK-228, or romidepsin, used for CTCL [54]. Romidepsin was initially isolated in 1994 from *Chromobacterium violaceum*, and although it displays minimal antibacterial properties, it was noted to have cytotoxic effects selectively on malignant cells [73]. In 2007, Karam et al. reported that in vitro treatment of bladder cancer cells with FK-228 caused tumour regression and also observed a similar finding in xenograft mice given intravenous doses of FK-228 [74]. To date, there have been two registered clinical trials using romidepsin in bladder cancer patients, one of which was terminated in 2006 due to poor accrual (NCT00087295). The other is currently active in phase 1 but has published dose-finding results, which showed that romidepsin is generally tolerable and has a similar toxicity profile to that of other HDAC inhibitors [60].

Valproic acid, a drug that is typically used to treat epilepsy and bipolar disorder, also has an inhibitory effect on HDACs. Given that valproate has been widely used for the management of neurological conditions, subsequent work on its HDAC-inhibiting properties focused on neurodegenerative disorders, such as Alzheimer’s disease, Huntington’s disease, and Parkinson’s disease [75]. It has also garnered attention in oncological settings, particularly in advanced cervical cancer where a double-blinded randomised controlled trial reported a significant improvement in PFS [76].

In urothelial cancer, multiple in vitro studies have shown that valproate is cytotoxic against human bladder cancer cells [77,78,79]. Intravesical instillation into rat bladder cancer models has also shown anti-tumorigenic properties, particularly when used in synergy with chemotherapy [79]. There have been two phase 1 clinical trials investigating valproate in bladder cancer patients, one of which is completed but without reported results (NCT01738815) and the other which is currently active (NCT01552434).

The last HDAC inhibitor involved in clinical trials for bladder cancer is entinostat. A 2020 in vitro study by Wang et al. showed that when used in combination with decitabine, a DNMT inhibitor, entinostat, exhibited a cytotoxic effect on chemoresistant bladder cancer cells without damaging normal urothelial cells [80]. A phase 2 trial of entinostat paired with pembrolizumab is currently underway, with results expected in late 2023 (NCT03978624).

### 3.5. Histone Methylation Readers

Several small-molecule BET inhibitors, including OTX015 and TEN-010, have progressed to clinical trials for haematological malignancy and select solid tumours, such as breast and prostate cancer [81]. Toxicity has been a major concern, however, and this is thought to be due to relatively poor selectivity. Further work into BET inhibition has yielded newer agents that target specific components of the bromodomain, such as ABBV-744, which acts primarily on bromodomain 2 [81]. While there have so far been no clinical trials using BET inhibitors in bladder cancer, a 2019 in vitro study by Li et al. reported the suppression of urothelial cancer cells by the BET inhibitor JQ1, an analogue of TEN-010 [82]. It has challenging pharmacokinetics due to its short half-life and poor bioavailability, however, which is perhaps a reason why it and so many other BET inhibitors have not progressed as far as other types of epigenetic agents.

## 4. Conclusions 

There are few treatment options for patients with advanced or metastatic bladder cancer who have failed first-line chemoimmunotherapy, making this an important area for future research. Epigenetic agents, long in use for haematological malignancies, have also shown promise for the treatment of advanced solid tumours. In vitro and in vivo studies have repeatedly shown that epigenetic agents can be of benefit for bladder cancer, even with chemoresistant cells, but so far, these results have not been convincingly replicated in human clinical trials.

A recurrent issue limiting the clinical applicability of epigenetic therapies has been a lack of selectivity for urothelial cancer cells, leading to high toxicity and a lack of demonstrable efficacy. Combining these agents with established first- or second-line therapy, such as platin-based chemotherapy or pembrolizumab, is a common strategy to improve the side-effect profile as it usually allows for a lower dose of epigenetic inhibitors to be given. This may also increase efficacy as many epigenetic inhibitors act in synergy with chemoimmunotherapy. However, most epigenetic drugs trialled for bladder cancer were originally developed for other diseases or malignancies, such as lymphoma, and greater selectivity for malignant urothelial cells remains the goal of treatment at this time. Although further work is required to identify novel epigenetic targets and agents specifically for urothelial cancer, it is inevitable that we will see progress in this domain and likely the greater applicability of epigenetic therapies to urothelial cancer of the bladder in the future. Urologists should be aware of this class of agent and understand their derivation and potential to improve bladder cancer-specific survival in the future.

## Figures and Tables

**Figure 1 cancers-15-01856-f001:**
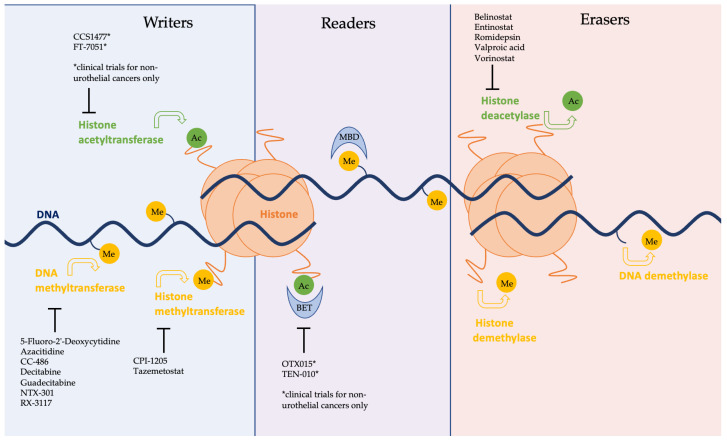
Schematic representation of the epigenome with enzymes grouped into three broad categories of ‘writers’, ‘readers’, and ‘erasers’. Methyl and acetyl groups are shown attached to DNA, and histones and are marked as ‘Me’ and ‘Ac’, respectively. Epigenetic inhibitors currently involved in clinical trials are listed in black, shown to be inhibiting their respective enzymes. Agents notated with an asterisk (*) are under investigation in clinical trials for non-urothelial cancers only.

**Table 1 cancers-15-01856-t001:** Summary of current human clinical trials investigating epigenetic inhibitors in bladder cancer.

Trial Identifier	Start Date	Expected End Date	Drug	Combination	Phase	Inclusion Cohort	Status	Results
DNMT inhibitors								
NCT04851834	25 August 2021	November 2023	NTX-301	Platinum-based chemotherapy	1/2	Locally advanced or metastatic bladder cancer; refractory/intolerant to standard of care therapies	Recruiting	Pending
NCT03179943	27 November 2017	July 2022	Guadecitabine	Atezolizumab	2	Advanced or metastatic urothelial carcinoma; must have received/been ineligible for CTx; must have had received PD-L1 or PD-1 targeting agent	Active—not recruiting	Pending
ISRCTN16332228	1 March 2016	10 July 2018	Guadecitabine	Cisplatin and gemcitabine	1b/2a	Incurable metastatic bladder cancer	Completed	Guadecitabine 20 mg/m^2^ is the recommended dose [55]
NCT00978250	20 August 2009	11 April 2019	5-Fluoro-2′-Deoxycytidine	Tetrahydrouridine	2	Advanced or metastatic urothelial carcinoma; received at least one line of standard therapy	Completed	Well-tolerate; AUC increase 4-fold; progression-free survival above expected [58]
NCT02030067	December 2013	July 2019	RX-3117	N/A (monotherapy)	1	Advanced bladder cancer	Completed	Not reported
NCT00030615	December 2001	September 2008	Decitabine	N/A (monotherapy)	1	Advanced or metastatic bladder cancer for which all other treatment has failed	Completed	Not reported
NCT02223052	27 October 2014	11 June 2018	CC-486 (oral form of azacitidine)	N/A (monotherapy)	1	Metastatic or inoperable bladder cancer	Completed	Not reported
NCT01478685	29 November 2011	17 November 2015	CC-486 (oral form of azacitidine)	Carboplatin or ABI-007	1	Relapsed or refractory urothelial carcinoma of the bladder, renal pelvis, ureter, or urethra	Completed	CC-486 is tolerated as a priming agent with carboplatin and ABI-007 [56]
NCT00005639	March 2000	July 2005	Azacitidine	Phenylbutyrate	1	Locally advanced or metastatic bladder cancer	Completed	Three doses were well-tolerated [57]
NCT02959437	27 February 2017	15 February 2019	Azacitidine	Pembrolizumab and epacadostat	1/2	Advanced or metastatic solid tumour, which has failed prior standard therapy	Terminated (by sponsors)	Not reported
EZH2 inhibitors								
NCT03854474	17 May 2019	30 June 2023	Tazemetostat (EPZ-6438)	Pembrolizumab	1/2	Locally advanced or metastatic urothelial carcinoma with progression during or following platinum-based CTx (or ineligible for CTx)	Recruiting	Pending
NCT03525795	14 December 2017	12 June 2019	CPI-1205	Ipilimumab	1/2	Unresectable or metastatic urothelial carcinoma (urethra, bladder, ureters, or renal pelvis)	Completed	Not reported
HDAC inhibitors								
NCT02619253	14 January 16	31 May 2023	Vorinostat	Pembrolizumab	1/2	Urothelial cell carcinoma—previously treated and progressive disease, locally advanced or metastatic; must have received a prior platinum-based regimen in the metastatic setting	Active, not recruiting	Pending
NCT00045006	July 2001	July 2008	Vorinostat	N/A (monotherapy)	1	Advanced or metastatic bladder cancer that is refractory to standard treatment	Completed	Not reported
NCT00565227	April 2007	November 2008	Vorinostat	Docetaxel	1	Bladder/urothelial cancer that has progressed after chemotherapy	Terminated (toxicity)	Not reported
NCT00363883	June 2006	December 2010	Vorinostat	N/A (monotherapy)	2	Bladder/urothelial TCC that has recurred or progressed on platinum-based CTx	Terminated (futility)	Limited efficacy and significant toxicity [59]
NCT05154994	14 January 2022	30 November 2023	Belinostat	Tremelimumab and durvalumab	1	Urothelial carcinoma with metastatic disease or with unresectable, locally advanced disease	Recruiting	Pending
NCT00413075	June 2006	August 2011	Belinostat	N/A (monotherapy)	1	Primary or metastatic solid tumour refractory to standard treatment	Completed	Not reported
NCT00413322	September 2005	March 2008	Belinostat	5-Fluorouracil	1	Advanced bladder cancer with progression on standard treatment	Completed	Not reported
NCT00421889	August 2005	February 2009	Belinostat	Carboplatin or paclitaxel	1/2	Urothelial carcinoma, received up to three CTx regimens in advanced disease setting	Completed	No published results; partial response in 4/15 patients
NCT01638533	12 June 2012	29 November 2018	Romidepsin	N/A (monotherapy)	1	Recurrent bladder cancer and concurrent hepatic impairment	Active, not recruiting	Similar toxicity to other HDAC inhibitors [60]
NCT00087295	June 2004	April 2006	FR901228 (Romidepsin)	N/A (monotherapy)	2	Metastatic or poorly differentiated TCC; progression after one CTx regimen	Terminated (poor accrual)	Not reported
NCT01552434	16 March 2012	31 March 2022	Valproic acid	Bevacizumab and temsirolimus	1	Metastatic urothelial cancer that is refractory to standard therapy	Active, not recruiting	Pending
NCT01738815	December 2011	May 2013	Valproic acid	N/A (monotherapy)	1	Suspected or confirmed bladder tumour	Completed	Not reported
NCT03978624	23 September 2020	1 October 2023	Entinostat	Pembrolizumab	2	MIBC ineligible for or refused neoadjuvant cisplatin-based CTx; pre-cystectomy	Recruiting	Pending

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
