# Peer review of "New Approaches to Targeting Epigenetic Regulation in Bladder Cancer"

_cancers, 2023, doi:10.3390/cancers15061856_

Round 1

Reviewer 1 Report

The authors introduced the epigenetic regulators and summarized the inhibitors of the regulators which have been proposed for bladder cancer therapy. The outline of the manuscript is straightforward, and each part is well-organized, especially the lists of the inhibitors, which are very detailed and informative. This work sheds light on the developmental stages of epigenetic inhibitors currently used in bladder cancer therapy, providing new ideas for bladder cancer treatments. Below are some questions/issues about the work.

Major issues:

Page 2, DNA methyltransferases part: Authors discussed the aberrant DNA methylation of multiple genes in urothelial carcinoma. Are there any literature reports of DNA methyltransferases or demethylases causing this hypermethylation and their relationship to bladder cancer?

Page 3, Erasers part: Authors should also include the DNA and histone demethylases for integrity.

Page 4, Readers part: Only proteins of BET family, which are the readers of histone methylation, were discussed. Authors should also introduce DNA methylation readers, like MBD-containing proteins (MBPs). Proteins in MBP family were also shown to be applied in bladder cancer treatment.

Minor issues:

Page 7 and Page 9: I would recommended replacing “Enhancer of Zeste 2 (EZH2)”, “CBP/p300”, “Bromodomain and extraterminal domain inhibitors (BETi)” with “Histone methyltransferases”, “Histone methylation readers” to maintain consistency. By the way, if unchanged, please use the full name of CBP/p300 for subtitle.

Reviewer 2 Report

Journal of Cancers

Review Article;

The article entitled “New approaches to targeting epigenetic regulation in bladder cancer”. The author investigate Epigenetics in bladder cancer. As opposed to genetic mutations in which the nucleotide sequence itself is altered, epigenetic alterations refer to changes to the genome that do not involve nucleotides. This is of great interest in cancer research because epigenetic alterations are reversible, making them a promising target for pharmacological agents. novel epigenetic therapies are important potential treatment options. The main classes of epigenetic inhibitors studied include DNA-methyltransferase (DNMT) inhibitors, histone deacetylase (HDAC) inhibitors, and enhancer of zeste homolog 2 (EZH2) inhibitors. At present, no phase 3 clinical trials have been registered. Though DNMT inhibitors have shown the most promise. Many patients with advanced or metastatic bladder cancer have limited treatment options, particularly when first- or second-line chemo immunotherapy fails. Epigenetic alterations, which are common in bladder cancer, are potential targets for drug therapies and these epigenetic agents are already in use for many cancers. While they have shown promise in pre-clinical trials for bladder cancer.

I carefully read the manuscript and found it is wonderful effort by the author to review epigenetic regulator in bladder cancer. But there is some minor revision needs and fulfill the mistake which the author could have done during writing. After minor revision, the article could be considered for publication in the prestigious Cancers Journal.

Comments for Authors

Ø  Abstract section, the author needs to revise the abstract. No need to mention the “Introduction, Material and Methods, Results and Conclusion” the author needs to revise and changed to plain text.

Ø  Write the keyword in alphabetical order.

Ø  “Introduction Section” the author needs to explain more the introduction section and include latest references.

Ø  During the manuscript line number 80 (1. Writer), line number 134 (2. Erasers), line number 157 (3. Readers) is completely out of understanding. The author needs to revise and remove such confusion subtitles.

Ø  It will be battered to add the combination effect with natural agents which show some promising effect.

Ø  It will be better if the author adds graphical image to present the mechanism of action at various pathway.

Ø  The author needs to revise the manuscript. There are many English mistakes.

Cite the following reference.

v  DOI: 10.2174/1871520622666220831124321
